# High Spatiotemporal Resolution PM2.5 Concentration Estimation with Machine Learning Algorithm: A Case Study for Wildfire in California

Qian Cui [1], Feng Zhang [2,3], Shaoyun Fu [4], Xiaoli Wei [4,*], Yue Ma [4] and Kun Wu [1]

[1] College of Atmospheric Sciences, Nanjing University of Information Science & Technology, Nanjing 210044, China; 20191201033@nuist.edu.cn (Q.C.); wukun@nuist.edu.cn (K.W.)

[2] Department of Atmospheric and Oceanic Sciences, Institute of Atmospheric Sciences, Fudan University, Shanghai 200433, China; fengzhang@fudan.edu.cn

[3] Shanghai Qi Zhi Institute, Shanghai 200232, China

[4] Jiading District Meteorological Bureau, Shanghai Meteorological Service, Shanghai 201800, China; shaoyunfu11@163.com (S.F.); mayue630@126.com (Y.M.)

\* Correspondence: xiaoliwei@knights.ucf.edu

**Abstract:** As an aggregate of suspended particulate matter in the air, atmospheric aerosols can affect the regional climate. With the help of satellite remote sensing technology to retrieve AOD (aerosol optical depth) on a global or regional scale, accurate estimation of PM2.5 concentration has become an important task to quantify the spatiotemporal distribution of AOD and PM2.5. However, due to the limitations of satellite platforms, sensors, and inversion algorithms, the spatiotemporal resolution of current major AOD products is still relatively low. Meanwhile, for the impact of cloud, the AOD products often have a serious data gap problem, which also objectively limits the spatiotemporal coverage of predicted PM2.5 concentration. Therefore, how to effectively improve the spatiotemporal resolution and coverage of PM2.5 concentration under the requisite accuracy is still a grand challenge. In this study, the fused high spatial-temporal resolution AOD data in our previous study were used to estimate the ground PM2.5 concentration through machine learning algorithms, the deep belief network (DBN). The PM2.5 data had spatiotemporal autocorrelation in geostatistics and followed the Gaussian kernel distribution. Hence, the autocorrelation model modified by Gaussian kernel function integrated with DBN algorithm, named Geoi-DBN, was used to estimate PM2.5 concentration. The cross-validation results showed that the Geoi-DBN ($R^2 = 0.86$, RMSE = 6.84 µg m$^{-3}$) performed better than the original DBN ($R^2 = 0.67$, RMSE = 10.46 µg m$^{-3}$). The final high quality PM2.5 concentration data can be applied for urban air quality monitoring and related PM2.5 exposure risk assessment such as wildfire.

**Keywords:** aerosol optical depth; PM2.5; data fusion; air quality; wildfire

## 1. Introduction

Atmospheric PM2.5 have many adverse impacts on climate change and human health. With an aerodynamic diameter less than or equal to 2.5 µm, it can directly reach the lungs and cause many related chronic diseases. Studies have found that long-term exposure to aerosol pollution can cause diabetes and cardiovascular and cerebrovascular diseases [1]. PM2.5 pollutants have become a general concern in many parts of the world. Although many countries have built PM2.5 ground monitoring networks that provide real time data, PM2.5 ground-based observations often cannot effectively characterize the distribution of PM2.5 pollution over a large area due to the sparse and spatially uneven distribution of ground monitoring sites. In addition, the high dynamic change of PM2.5 pollutants such as smoke and dust will generate and emit a large amount of particulate matter in a short time, with heterogeneity characteristics in spatial distribution. Consequently, accurate and

seamless mapping of PM2.5 concentration with high spatiotemporal resolution by using satellite remote sensing data with the help of a machine learning approach is critically important.

Heterogeneous PM2.5 is caused by the emission of PM2.5, spatial distribution of buildings in ground surface, changes of the intensity of human activities in regional environments, and many other factors [2]. In the early days of remote sensing, researchers usually used a simple linear model to estimate the ground PM2.5 concentrations from satellite AOD (aerosol optical depth) data, without fully considering the covariant mechanism between satellite AOD and ground PM2.5 data [3,4]. Further research found that the statistical relationship between PM2.5 concentration and AOD was greatly affected by factors such as aerosol type, vertical distribution characteristics, and the moisture absorption effect of particles as well as spatiotemporal anisotropy between PM2.5 and AOD [5–9]. Hence, the relative humidity and boundary layer height have been introduced into the PM2.5 estimation model to improve the accuracy. The relationship between AOD and PM2.5 is complicated and it is obvious that the uncertainty of the estimated PM2.5 concentration is large when using one single model [10]. Furthermore, with the deepening understanding of the physical and chemical characteristics and formation mechanism of PM2.5, the number of factors and amount of data involved in PM2.5 estimation have increased rapidly. Models developed from general linear regression to complex multiple regression such as the mixed-effects model, geographically weighted regression model, land use regression model, and other empirical statistical models, etc. [11–15]. Generally, the relationship between AOD and PM2.5 concentration is non-linear [16]. All traditional empirical models are unable to optimally express this nonlinear relationship. Therefore, machine learning methodology was brought into PM2.5 concentration estimation, which has a strong learning ability and can effectively establish the complex nonlinear relationship between PM2.5 concentration and its influencing factors. Studies have found that the PM2.5 estimated accuracy of machine learning is better than traditional linear and semi empirical statistical models [17].

AOD is positively correlated with PM2.5 [18,19]. Aerosol optical depth (AOD) products observed from multi-source satellite such as MODIS (Moderate-resolution Imaging Spectroradiometer), VIIRS (Visible Infrared Imaging Radiometer Suite), and GOES (Geostationary Operational Environmental Satellite) provide the chance to obtain fine resolution PM2.5 maps. However, the AOD product of a single instrument usually has limited application on a large-scale due to low coverage and the amount of missing data caused by cloud contamination and bright surfaces. Furthermore, system bias of sensors and different inversion algorithms make it hard to provide a consistent AOD dataset from different satellites. Most AOD products have a low resolution, which make it difficult to meet the research requirements of small-scale areas such as individual urban area [20,21]. For example, MODIS flown on the Terra and Aqua satellites can provide AOD products in three spatial resolutions: 10 km, 3 km, and 1 km. The VIIRS, as a substituted sensor of MODIS, has a 750 m AOD product. In temporal resolution, most AOD data observed by polar-orbiting satellites are collected daily. In contrast, geostationary satellites can provide minute-scale AOD data with lower spatial resolution such as GOES-16 (4 km/15 min) and Himawari-8 (5 km/10 min) [22]. Compared to the observed AOD data, atmospheric chemistry model simulated AOD data such as the MERRA-2 AOD data are global-coverage, low accuracy, and coarse resolution for the complex structure of model and vast issues considered in simulation [23–25]. Overall, no single source of AOD data can provide high precision, high resolution, and high coverage AOD data, which will result in PM2.5 concentration estimation with the same defects.

Fusing multi-source heterogeneous AOD data is an effective way to address the limitations of a single source of AOD data and generate higher quality AOD data [26]. Previous studies that have researched AOD data fusion algorithms have been mainly confined to different areas and by different sources of instruments. Yang and Hu (2018) applied a spatiotemporal kriging approach to fill in the gaps in the MODIS AOD product [27]. Spatial statistical approaches showed that improved data coverage is subject to unsatisfactory

results when the available data are sparsely distributed. In light of this, combining aerosol products from multiple sensors has been proposed [28]. Tang et al. (2016) reported on the BME method to fuse missing AOD with MODIS and SeaWiFS (Sea-viewing Wide Field-of-view Sensor) (13.5 km) products [26]. Sogacheva et al. (2020) fused 15 different AOD products together and obtained a consistent monthly AOD dataset by using an algorithm of AOD weighted with the results of the AERONET (AErosol RObotic NETwork) verification [29]. However, all of these studies have failed to capture non-randomness in missing AOD values. Xiao et al. (2017) further combined satellite AOD with model simulations by using a multiple imputation method [30]. It remains difficult to obtain accurate aerosol measurements for the uncertainties and coarse spatial resolution brought by the models. Therefore, our recent studies have addressed these limitations by proposing a data fusion algorithm, MQQA-BME, which is an integrative algorithm by synergizing the advantages of MQQA (Modified Quantile-Quantile Adjustment) and BME (Bayesian Maximum Entropy). In the MQQA-BME algorithm, MQQA is a complementary tool for adjusting the systematic biases in the data sources and BME is used for data downscaling and prediction [31,32]. The integrated MQQA-BME algorithm has been proven to perform well for dynamic multisource data fusion such as TOA reflectance and AOD data.

Based on the fused AOD data, it is possible to seamlessly map the ground PM2.5 concentration with fine spatiotemporal resolution, full coverage, and precision. Spatiotemporal information is essential in the estimation of ground PM2.5 concentration. Li et al. (2016) introduced spatiotemporal information into the DBN (deep belief network) model and called it the Geoi-DBN model, which achieved satisfactory results in ground PM2.5 estimation [33]. However, it is based on the traditional Moran index to describe the spatiotemporal information by using the weight of the correlation, which is a linear statistical relationship and cannot accurately depict the real distribution of PM2.5. Later, several researchers further improved this algorithm based on a random forest model [34] and extreme random tree model to map the ground PM2.5 concentration [35]. However, the majority of these studies generally defined a specific window as the spatiotemporal impact domain, which is prone to subjective factors. In this context, the deep confidence network algorithm is used to mine the complex nonlinear relationship between PM2.5 concentration and AOD data. Meteorological factors at high resolution are also considered when building the estimated model since they are known to have an influence on the PM2.5 concentration. Additionally, drawing on the concept of spatiotemporal autocorrelation in geostatistics, the Gaussian kernel model was brought into the deep confidence network, which will improve the accuracy of PM2.5 concentration by considering prior and neighborhood information. The scale of the window is automatically defined by the spatiotemporal covariance model, which reflects the autocorrelation of data in the spatiotemporal domain.

The science questions to be answered in this paper include: (1) Can fused AOD via multisource data better estimated PM2.5 concentration with high spatiotemporal resolution, especially in a highly dynamic forest fire event? and (2) can the Geoi-DBN algorithm significantly improve the modeling accuracy by incorporating the Gaussian kernel and spatiotemporal covariance models? Given the results of our previous studies, the MQQA-BME algorithm can produce better quality AOD data/products. The objectives of this paper are thus to: (1) evaluate the gap filling and spatiotemporal resolution, improving capabilities in estimating ground PM2.5 concentration of fused AOD data, and (2) demonstrate the Gaussian kernel model and Gaussian kernel model integrated with the DBN algorithm with the aid of meteorology factors to improve the estimation accuracy of ground PM2.5 concentration.

## 2. Study Area and Datasets

The study area comprises central and southern California, as shown in Figure 1. The hourly average observed ground PM2.5 concentration data from September to November in 2018 (fall) was collected from the U.S. EPA (Environmental Protection Agency) website (https://www.epa.gov/outdoor-air-quality-data (accessed on 6 January 2021)). Twenty-

three stations were set up for air pollution monitoring in our study area (Figure 1). The fused AOD data obtained from our previous study [31,32] had a 1 km spatial resolution and a temporal resolution of a half hour. The fused AOD data were derived from the MERRA-2 (Modern-Era Retrospective analysis for Research and Applications, Version 2) reanalysis AOD data, the geostationary satellite GOES-16 (Geostationary Operational Environmental Satellites 16) AOD product, and polar-orbiting satellite Terra MODIS AOD data (MCD19A2/AOD, inversion by MAIAC (Multi-Angle Implementation of Atmospheric Correction) algorithm). We obtained meteorological variables from the ECWMF (European Medium-Term Weather Forecast Center) (https://www.ecmwf.int/en/research/climate-reanalysis (accessed on 6 January 2021)) including wind speed at 10 m above ground (WDSP, m/s); air temperature at 2 m above ground (TMP, K), relative humidity (RH, %), surface pressure (PRESS, Pa), total precipitation (TP, mm), and planetary boundary layer height (PBLH, m). These meteorological data have a spatial resolution of $0.25° \times 0.25°$ and temporal resolution of one hour. In this study, we used the 00:00 UTC, 06:00 UTC, 12:00 UTC, and 18:00 UTC data as examples and the meteorology data were downscaled to AOD spatial resolution (1 km) by using the nearest neighbor interpolation.

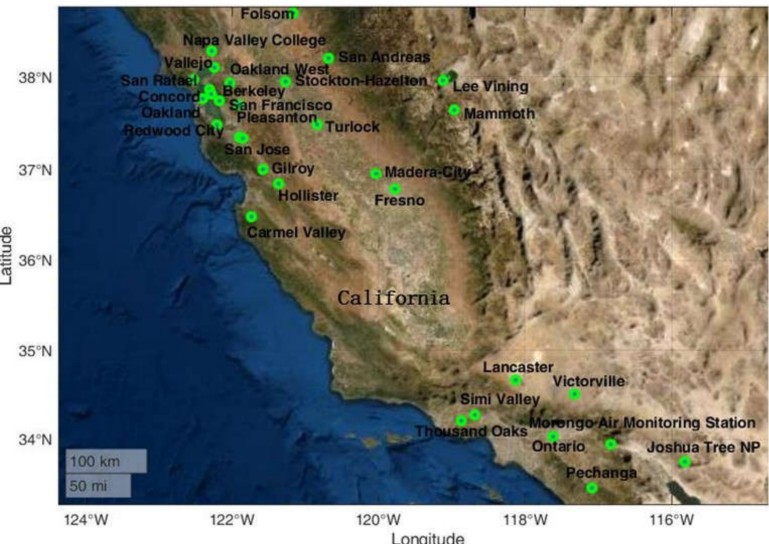

**Figure 1.** The study area of central and southern California as well as the locations of the PM2.5 stations.

## 3. Methods

### 3.1. Integration of the MQQA-BME Algorithm

The MQQA algorithm aims to address the systematic bias among the data information of multiple sources by using the quantile–quantile mapping theory to increase the comparability of multiple sources of data information. Since the theoretical root of the MQQA algorithm is matching based on probability distribution, the revisions for the more random deviations were not significant. At the same time, MQQA could only perform error correction on the existing image element values, and could not realize the reconstruction operation of missing data. BME of a nonlinear fusion algorithm could combine physical prior knowledge with probability theory for prediction [36] and perform better in downscaling with high dynamic parameters [37,38]. In addition, considering the autocorrelation of spatiotemporal neighborhood data, we could interpolate to fill in the missing data, and thus significantly improve the data coverage. However, the BME algorithm for downscaling fusion in using simultaneous observations from multiple satellites does not consider the contribution of long-time series of historical information and the problem of systematic bias between sensors. The MQQ-BME algorithm combines the advantages of these two algorithms and overcomes the defects of the original algorithm by introducing the error model constructed by the MQQA algorithm based on historical data information into the Bayesian maximum information entropy algorithm [31]. So far, the integrated MQQA-BME algo-

rithm has been successfully applied to multi-source heterogeneous AOD data fusion [32], providing a data basis for seamless mapping of ground-level PM2.5 concentration. The MQQA-BME algorithm is described in detail in our precious study [31].

The BME theory can integrate physical knowledge with probability law as a nonlinear estimator for prediction that has been widely used in atmospheric studies [37]. Generally, the physical knowledge can be divided into two categories: general knowledge and specific knowledge. The general knowledge expresses the global characteristics such as consistent pattern, which is described by the mean trend value as well as the spatial and temporal dependence indicated by its covariance. The datasets include a hard dataset and a soft dataset. The hard data are deemed accurate with good quality. The soft data are usually data with uncertainty and data missing. In BME theory, a Gaussian process error model is derived from the discrepancies between hard data and soft data to help explore the error propagation in between the two adjacent hard data. Obviously, the Gaussian process error model is not a physical model for bias correction during prediction. For this reason, we improved the Gaussian process error model in this analysis by introducing the MQQA into the BME for systematic bias correction and removing the systematic bias from different data sources to reduce the error propagation. By integrating MQQA and BME with the Shannon information in a format of PDF (probability density function) [38], the MQQA-BME approach can obtain high posterior information about the spatiotemporal structure under estimation to formalize the specific knowledge. Finally, the estimated mean value and variance data are obtained at an estimated point via a mathematical optimization process to achieve data fusion.

### 3.2. Deep Belief Network Algorithm (DBN)

DBN is a generative deep neural network model [39]. It is composed of multiple hidden layers that can operate by the RBM (restricted Botzman machine), which has two layers (a single visible layer and a hidden layer) of feature-detecting units [39]. There are connections between layers, but not between cells within layers, and hidden layer cells are trained to capture the correlations of higher-order data represented in the visible layer. In other words, several RBMS are connected in series to form a DBN, where the hidden layer of the previous RBM is the explicit layer of the next RBM, and the output of the previous RBM is the input of the next RBM [40]. The architecture of the DBN network is shown in Figure 2.

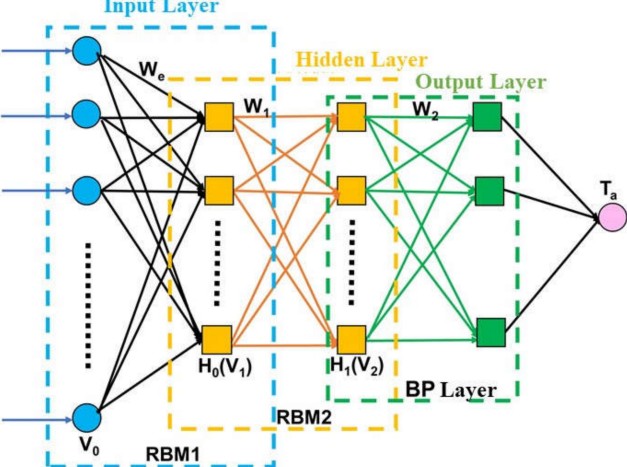

**Figure 2.** Structure of the DBN model.

The nodes of RBM layers and number of central neural layers are key parameters of the DBN model. Generally, the more RBM layers there are, the better the ability to simulate complex nonlinear relationships, however, too many layers can lead to overfitting [33,41]. In addition, a previous study showed that the number of central neurons determined by the

number of input variables ($n$) and output variables ($u$), ranges from $2\sqrt{n} + u\ to\ 2n + 1$ [42]. In the present study, given that this is a complex nonlinear atmospheric environment, it is advantageous to set the layers of RBM as two layers and the number of central neural layers as 20 per layer, according to the input and output to improve the computational speed. The parameters that take part in the DBN model to estimate PM2.5 are as follows (refers to the nonlinear DBN estimation model):

$$PM2.5 = f\ (MERRA\ 2\_GOES\ 16\_MAIAC\ Fused\ AOD, Lat, Lon, DOY, U10,\ V10, TMP, PBLH, \\ PRESS, RH, TP) \tag{1}$$

where *Lat* is the latitude; *Lon* is the longitude; *DOY* is the Day of Year; *U*10 is the east–west component of the wind vector; and *V*10 is the north–south component of the wind vector. *TMP* is the air temperature at 2 m above ground; *RH* is the relative humidity; *PRESS* is the surface pressure; *TP* is the total precipitation; and *PBLH* is the planetary boundary layer height.

### 3.3. Geoi-Deep Belief Network (Geoi-DBN)

Li et al. (2017) found that the accuracy of PM2.5 estimation can be improved by using the DBN model with consideration of the spatiotemporal information of neighborhoods [33]. Wei et al. (2020) further compared and analyzed various deep learning algorithms and found that Geoi-DBN performed better than others, with an $R^2$ of 0.88 [31]. When estimating PM2.5 data of a single site, PM2.5 data of the site had good autocorrelation with the surrounding data in a certain range since all of the data belonged to the same emission source. Moreover, there was an obvious time dependence between the data of one day and the data of adjacent time [43]. By considering the spatiotemporal autocorrelation at model initiation, some researchers have used the Moran I index with a spatial weight matrix to explore the spatiotemporal autocorrelation [33]. However, the actual spatial distribution of ground PM2.5 concentration cannot be well expressed by simply introducing Moran I into the estimation model [33]. Therefore, Gaussian kernel function was introduced into this study to describe the real spatiotemporal distribution of PM2.5 [34], as shown in Equations (2)–(5):

$$S\_PM2.5 = \frac{\sum_{i=1}^{n} WS_i PM2.5_i}{\sum_{i=1}^{n} WS_i} \quad i = 1, 2, \ldots. n \tag{2}$$

$$T\_PM2.5 = \frac{\sum_{j=1}^{m} Wt_i PM2.5_j}{\sum_{j=1}^{n} Wt_j} \quad j = 1, 2, \ldots. m \tag{3}$$

$$WS_i = exp\left(\frac{-d^2}{2var(PM2.5)_{window\_range}}\right) \tag{4}$$

$$WT_j = exp\left(\frac{-t^2}{2var(PM2.5)_{window\_range}}\right) \tag{5}$$

where *S_PM*2.5 and *T_PM*2.5 are the autocorrelation of PM2.5 concentrations in the spatial and temporal neighborhoods, respectively; d is the Euclidean distance between the PM2.5 observations and the point to be estimated within the autocorrelation window; *var(PM2.5)$_{window\_range}$* is the variance of the data within the autocorrelation window; $WS_i$ is the spatial weight relationship between the *i*-th PM2.5 data within the autocorrelation window size; and $WT_j$ is the temporal weight relationship between the *j*-th PM2.5 data within the temporal window size.

Additionally, the window size depicting the autocorrelation impacts between the estimation data and neighborhood data was very significant. Because the data at a close range are highly correlated, small window size will lead to over-fitting of the PM2.5 estimation. In contrast, a large window may introduce non-homologous data to the calculation, which would reduce the accuracy of PM2.5 estimation. The use of an artificial constant attempts to obtain the optimum window size, which is a popular method to obtain

the accuracy result, the optimum parameters were determined by conducting parameter sensitivity experiments. Multiple attempts are tedious, and it is difficult to avoid the subjective influence. To solve this problem, a spatiotemporal covariance model was brought into this study, automatically determining the window size and better characterizing the spatial distribution of PM2.5 concentrations. As mentioned previously, the spatiotemporal covariance model was the core of the spatiotemporal correlation model, which should portray the spatiotemporal correlations based on the interdependence between the temporal parameters and the spatial parameters in the autocorrelation window domain [44].

Therefore, through analysis of the observation data of PM2.5 in the long-term series in the study area based on the longitude, latitude, and time (Figure 3), respectively, it was found that by exploring the least square method, the covariance relationships in the spatiotemporal domain followed an exponential distribution. The autocorrelation coefficient between PM2.5 data decreased exponentially with increasing distance. It is noteworthy that the pattern in the temporal domain was more complicated with perhaps some sort of fluctuation after a period of time. This may have been caused by the high dynamic changing and heterogeneous characteristics of PM2.5 concentration in the spatiotemporal domain [45]. Therefore, the temporal distribution needs further study in the future. Here, we still used an exponential distribution to describe a large scale trend in time. In Figure 3, the spatial distance reached three pixels and the temporal lag was larger than three days, the covariance value tended toward 0, meaning that the correlation disappeared. In this study, a $1.5 \times 1.5 \times 3$ autocorrelation window size was used. Finally, the optimized Geoi-DBN model is described in Equation (6):

$$PM2.5 = f(MERRA\ 2\_GOES\ 16\_MAIAC\ Fused\ AOD, Lat, Lon, DOY, U10, \ V10, TMP, PBLH, \\ PRESS, RH, TP, S\_PM_{2.5}, T\_PM_{2.5}) \tag{6}$$

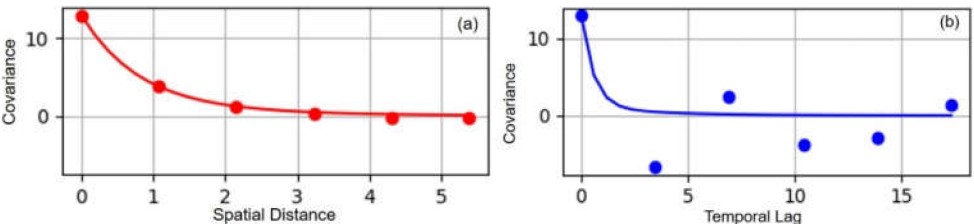

**Figure 3.** (**a**,**b**) Spatial and temporal covariance models for the PM2.5 data.

Finally, the efficacy of such validation can be confirmed by a set of statistical indexes through Equations (7)–(9).

The root-mean-square error (*RMSE*) is:

$$RMSE = \sqrt{\frac{1}{n-1} \sum_{i=1}^{i=n} (y_i - \hat{y}_i)^2} \tag{7}$$

The coefficient of determination ($R^2$) is:

$$R^2(\hat{y}_i, y_i) = 1 - \frac{\sum_{i=1}^{i=n} (f_i - \hat{y}_i)^2}{\sum_{i=1}^{i=n} (y_i - \hat{y}_i)^2} \tag{8}$$

The mean absolute difference (*MAD*) is:

$$MAD = \sum_{i=1}^{n} \frac{|\hat{y}_i - y_i|}{n} \tag{9}$$

where $y_i$ and $\hat{y}_i$ are the predicted and ground truth data, respectively. $n$ is the total number of pair wised samples and $f_i$ is the fitted (or modeled, or predicted) value $f1, ..., fn$ of $y_i$.

## 4. Results

### 4.1. Results of Multi-Source Heterogeneous AOD Fusion

Table 1 indicates the comparison between AERONET AOD data and MERRA-2 AOD, GOES-16 AOD, MAIAC AOD, and MERRA-2_GOES-16 fused AOD as well as MERRA-2_GOES-16_MAIAC AOD fused data, respectively, in our study area in the fall of 2018. The results show that the $R^2$ ranged from 0.34 to 0.53, with the fused $R^2$ of 0.48 being acceptable due to gap filling. Among the products of multisource AOD data, the AOD products of polar-orbiting satellites have the highest accuracy (MAIAC), which is also consistent with the previous studies of many researchers [46]. MERRA-2 data in the low-value region correlated well with ground data, but there was an underestimation, possibly due to missing emission source inventories in the aerosol model [47]. In comparison, the AOD data generated by MERRA-2 had a higher accuracy than GOES-16 data and an advantage of full coverage image with a sample size of 674. Nevertheless, GOES 16 is still recommended as one of the fusion sources benefit because of its high temporal resolution of 30 min. The intermediate fusion data of MERRA-2 and GOES 16, namely MERRA-2_GOES-16 AOD and the final fusion data of MERRA-2, GOES 16, and MAIAC AOD data, called MERRA-2_GOES-16_MAIAC AOD, were also verified. Figure 4 shows the distribution of R2, MAD, and RMSE between the MERRA-2_GOES-16_MAIAC AOD and AERONET observed AOD. Obviously, AERONET AOD sites located in coastal locations could result in large errors for sites close to the coast in California where it is easily influenced by the marine environment [48]. Overall, the fusion accuracy of MERRA-2_GOES-16_MAIAC AOD data was slightly lower than the MAIAC AOD data, but the spatial coverage and temporal resolution were greatly improved. Moreover, the final fused MERRA-2_GOES-16_MAIAC AOD data were in good agreement with the ground-based AOD data.

**Table 1.** The accuracy result of various AOD data.

| AOD | $R^2$ | RMSE | MAD | N |
|---|---|---|---|---|
| MERRA-2 | 0.41 | 0.10 | 0.05 | 488 |
| GOES 16 | 0.34 | 0.11 | 0.07 | 430 |
| MERRA-2_GOES 16 | 0.30 | 0.14 | 0.10 | 674 |
| MAIAC AOD | 0.53 | 0.07 | 0.04 | 392 |
| Final Fused AOD | 0.48 | 0.08 | 0.05 | 674 |

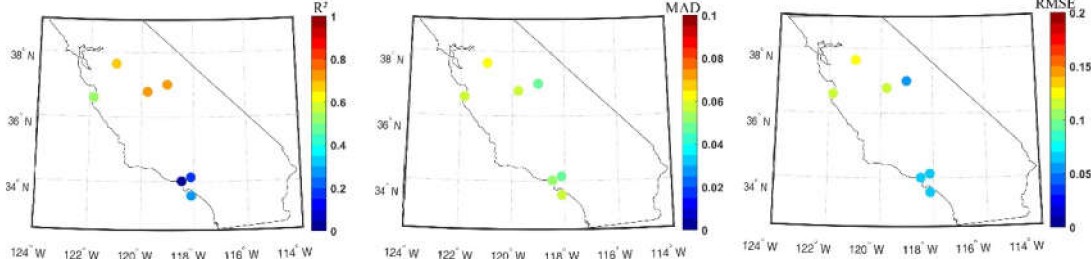

**Figure 4.** Spatial distribution of $R^2$, MAD, RMSE of final fused AOD with AERONET observed AOD data.

### 4.2. Potential Effects of Variables on PM2.5

AOD had a good spatial and temporal correlation with PM2.5, especially when pollutants were concentrated in the lower atmosphere. AOD is highly positively related to PM2.5, whereby the correlation coefficient can even reach above 0.75 in some regions [49]. A large number of studies have found significant differences in aerosol chemical composition and regional meteorological fields in different regions, which may lead to the weakening of this relationship in some regions or even negative correlation, limiting the application of estimation model.

It is difficult to observe the properties of aerosols for the complex atmospheric conditions in a certain area [50,51]. To achieve accurate statistical analysis of the relationship between AOD and PM2.5, several parameters are necessary to introduce into the estimation model. Theoretically, PM2.5 concentrations are closely related to near-surface meteorological factors such as wind speed, wind direction, and relative humidity [52–54]. For instance, the increase in relative humidity promotes the hygroscopic growth of pollutants and dust-haze [55]. It was found that when the relative humidity reached 98–99%, the optical characterization of AOD changed by 25%, while, when the relative humidity reached 50–80%, the optical characterization of AOD only changed by 5% [56]. Concurrently, the turbulent processes in the atmospheric boundary layer play an important role in the diffusion and dilution of pollutants. When the turbulence activity decreases, the height of mixing layer decreases and the atmospheric stability increases. Pollutants and water vapor are concentrated in the boundary layer, resulting in a large amount of aerosol accumulation [57]. Hoff revised the formula of AOD estimated PM2.5 by considering the relative humidity and atmospheric boundary layer height impacts as follows [6]:

$$\tau = PM2.5H\,f(RH)\frac{3Q_{ext,dry}}{4\rho r_{eff}} \tag{10}$$

where $H$ is the uniformly mixed atmospheric planetary boundary layer height; $\tau$ is the aerosol optical thickness; $f(RH)$ is the ratio of ambient extinction coefficient to dry extinction coefficient; $\rho$ is the mass density function of aerosol (g m$^{-3}$), $Q_{ext,dry}$ is the extinction coefficient of Mie scattering and the corresponding radius of the particles. *reff* is the area weighted average radius.

It should be pointed out that quantitative analysis of the statistical relationships between input parameters of machine learning models and PM2.5 concentrations in the study area is necessary and will screen out effective variables [58]. Since machine learning models are "black boxes", it is difficult to account for the estimation results with materialization mechanisms or mathematical formulas. Currently, it is impossible to improve machine learning models based on the relevant materialization mechanisms for better understanding and improving PM2.5 estimation. To overcome this limitation, statistical analysis of the input parameters before estimation can help deepen the understanding of the materialization mechanisms between PM2.5 and other parameters.

Figures 5 and 6 show the diurnal mean distributions of EPA observed PM2.5 concentration and AERONET AOD data in the fall of 2018 in our study area. As seen from the figures, there were two peaks at around 15:00–18:00 UTC and 00:00–03:00 UTC before the wildfire outbreak, consistent with the diurnal pattern revealed in previous studies [59]. The AOD data and PM2.5 data had a synchronous change. Both reflected high values due to wildfire outbreaks, which emit a lot of pollutants by biomass combustion. Therefore, in this study, AOD was applied as the major factor to estimate ground-level PM2.5 concentration. During the wildfire period, note that there were two peaks in the AOD data while there was only one peak for PM2.5 later in the day. This is because PM2.5 is not only related to AOD, but is also influenced by meteorological factors [45]. As shown in Figure 5, PM2.5 data are more constant during the wildfire period than AOD. The deeper reason is that it had a data gap of AOD around the fire point and some smoke plume is also easily treated as cloud. The influence of smoke plume and heterogeneous surface lead to the bias of satellite AOD retrieval. The physicochemical process of connect uplift smoke plume to surface concentrations needs further study in the future.

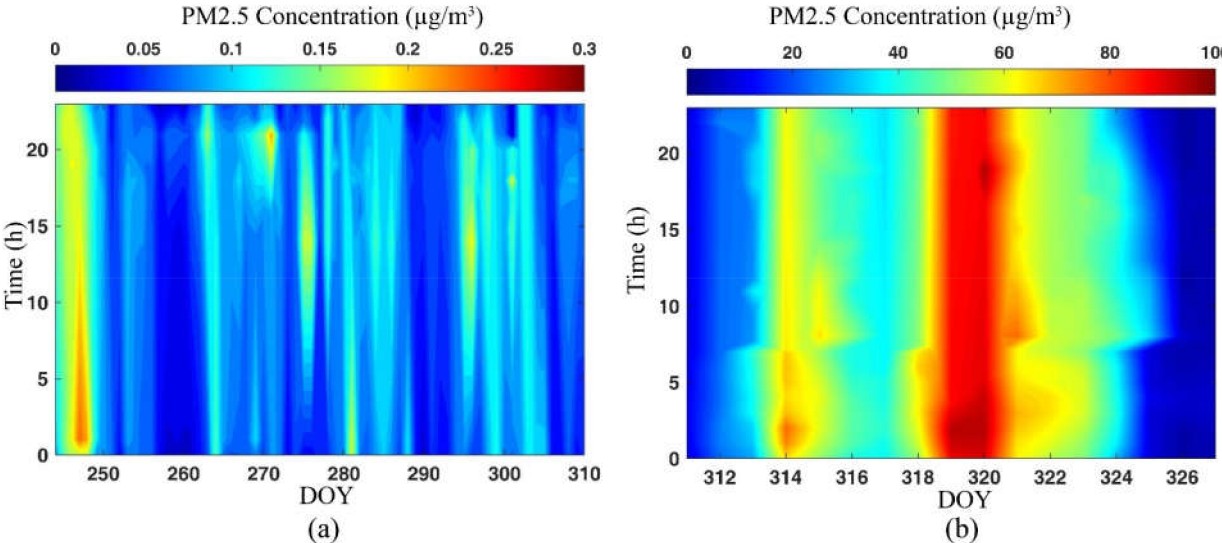

**Figure 5.** Daily and hourly distribution of the observed ground PM2.5 concentration based (**a**) before wildfire outbreak; (**b**) during the wildfire outbreak (two peaks time at 15:00–18:00 UTC and 00:00–05:00 UTC).

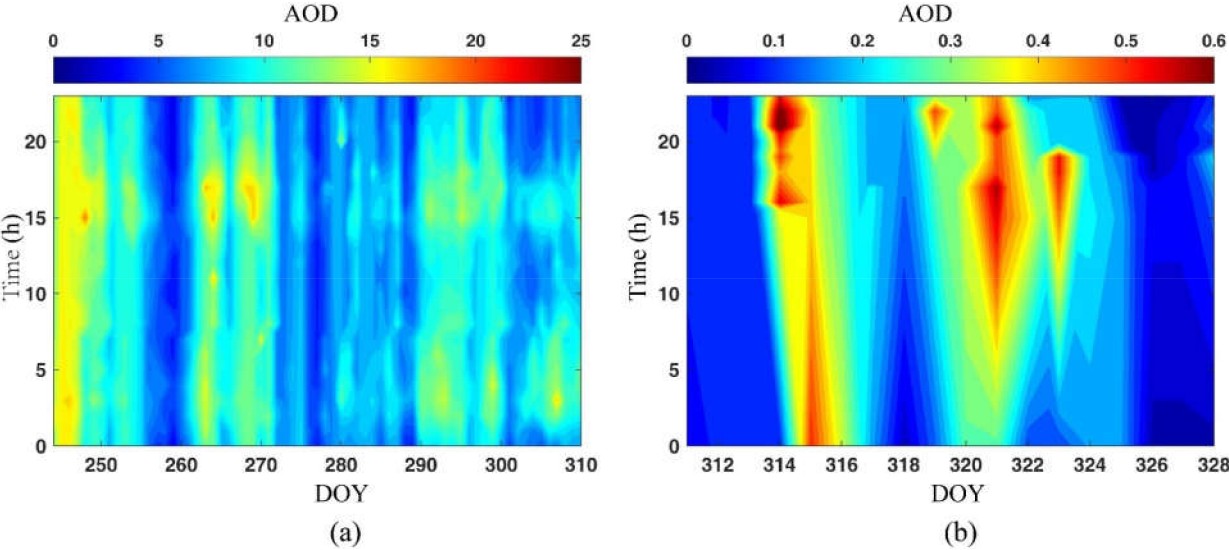

**Figure 6.** Daily and hourly distribution of observed AOD data of AERONET sites (**a**) before wildfire outbreak; (**b**) during the wildfire outbreak.

In order to further investigate the quantitative relationship between AOD and ground PM2.5 concentration, Figure 7a,b show the annual mean AOD and PM2.5 spatial distribution of AERONET station and ground EPA stations, respectively. Results indicate that the spatial distribution of the two, with high values in the northwestern region and lower values in the southern region, is consistent with the transport of smoke from wildfire according to the HYSPLT (Hybrid Single Particle Lagrangian Integrated Trajectory) online model (https://www.ready.noaa.gov/HYSPLIT.php (accessed on 10 March 2021)). The HYSPLIT forward trajectories in Figure 8 show that the polluted substance transfers out to the center of California from 11 to 25 November 2018 were mainly from the north. The surrounding area where biomass has been burning usually indicates the increase in PM2.5 concentration.

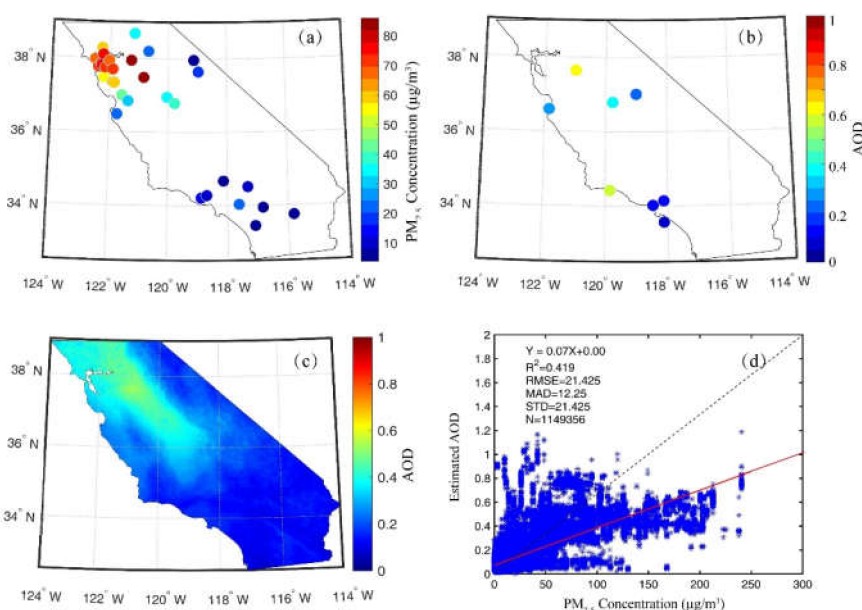

**Figure 7.** The comparison of AOD and PM2.5 during the wildfire period. (**a**) Distribution of daily mean PM2.5 concentration at EPA sites; (**b**) Daily mean AOD data of AERONET sites; (**c**) Spatial distribution map of AOD data by MAIAC AOD data; (**d**) Scatter plot of ground PM2.5 concentration vs. fused AOD data.

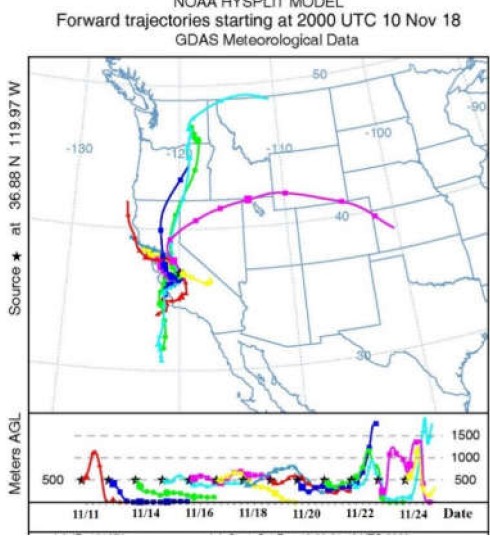

**Figure 8.** Fifteen days forward trajectories starting on 11 November 2018.

Figure 7c is the AOD distribution from a satellite point of view, which also follows the spatial distribution of ground stations. Figure 7d further verifies that there is a high correlation between PM2.5 and AOD in this study area ($R^2$ = 0.42). Overall, it is feasible to use AOD satellite observation data and machine learning algorithms to estimate the ground-surface PM2.5 concentration in this study area.

### 4.3. High-Resolution PM2.5 Concentration Estimation Based on AOD Fusion Products

Figure 9 compares and analyzes the accuracy of both DBN and Geoi-DBN algorithms in estimating the ground-level PM2.5 concentrations during the California wildfire period in the fall of 2018. The results for the original DBN and Geoi-DBN algorithm by 10-fold cross-validation between the measured and predicted PM2.5 concentrations from the DBN model and Geoi-DBN model had an $R^2$ of 0.67 and 0.86 and RMSE of 10.46 µg m$^{-3}$ and 6.84 µg m$^{-3}$, respectively. We found that the accuracy of the Geoi-DBN algorithm was

significantly better than the original DBN algorithm. This indicates that the Geoi-DBN algorithm could estimate the ground PM2.5 concentration more accurately to introduce the a priori information of the observation of the PM2.5 concentrations of surrounding stations. Estimated PM2.5 concentrations were more correlated to the PM2.5 concentration in the spatiotemporal domain associated with S_PM2.5 and T_PM2.5 variables. Figure 9b,d show the estimated spatial distribution of PM2.5 concentrations, and both could estimate the PM2.5 distribution well since they both belong to the DBN algorithm framework. The PM2.5 concentrations were significantly higher in the north and lower in the south, and the spatial distribution of estimated PM2.5 concentrations from two algorithms was consistent with the ground station observation data, ranging from 0 μg m$^{-3}$ to 80 μg m$^{-3}$. However, the original DBN algorithm, to some extent, had overestimation in the high value area and its slope was lower than Geoi-DBN as it contains a number of underestimated valuations. This phenomenon needs further research in the future.

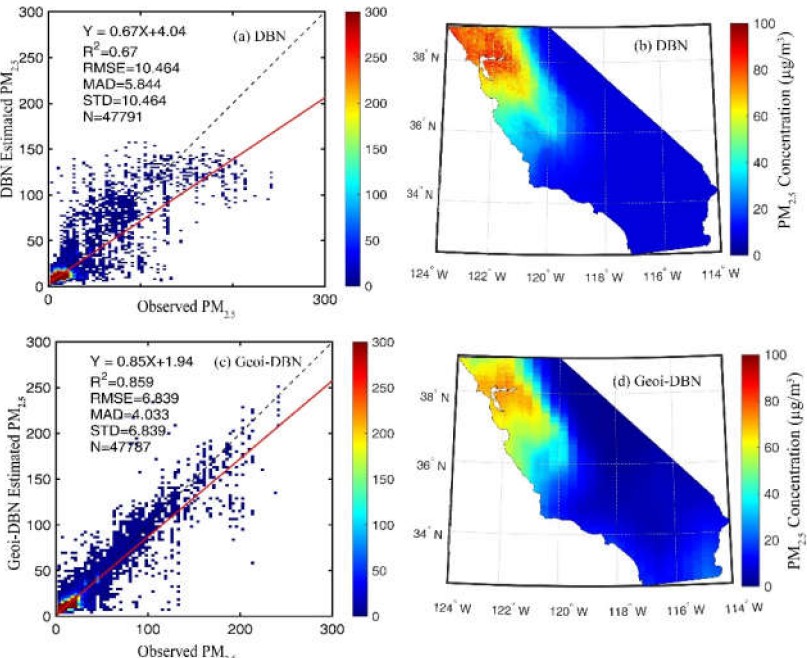

**Figure 9.** Comparison between the DBN model and Geoi-DBN model in estimating PM2.5 concentration during the wildfire period in 2018: (**a**) Scatter plot of cross-validation of DBN model; (**b**) Spatial distribution of estimated PM2.5 concentration by DBN model; (**c**) Scatter plot of cross-validation of the Geoi-DBN model; (**d**) spatial distribution of thee estimated PM2.5 concentration by the Geoi-DBN model.

Furthermore, when catastrophic air pollution events such as wildfire occur, the PM2.5 concentration observation is often required to be of high resolution, in order to provide timely and accurate services for disaster prevention and mitigation. During the two peak periods of hourly PM2.5 concentration, as shown in Figures 5 and 6, to further validate the performance of the Geoi-DBN algorithm, the PM2.5 concentrations point data at the ground-based sites of the observed data and the Geoi-DBN algorithm estimated data on 10 November 2018 (the breakout of wildfire), 16 November 2018 (the developing of wildfire), and 25 November 2018 (the end of wildfire) are shown, respectively (Figures 10 and 11), and the associated grid spatial distribution map of the estimated PM2.5 concentration is also shown in these figures. At the time of the wildfire, the value continued to increase in the high value region, but the high value range decreased. Ten sites in the north reached 100 μg m$^{-3}$, pinpointing the serious PM2.5 exposure risk. Since then, as the disaster was brought under control, the concentration and area of PM2.5 dropped significantly to lower than 20 μg m$^{-3}$. This further demonstrates that the estimated high spatial and temporal

resolution PM2.5 concentration (1 km and hourly) by using the Geoi-DBN algorithm with AOD fused data can better represent the variation of ground PM2.5 data during wildfire.

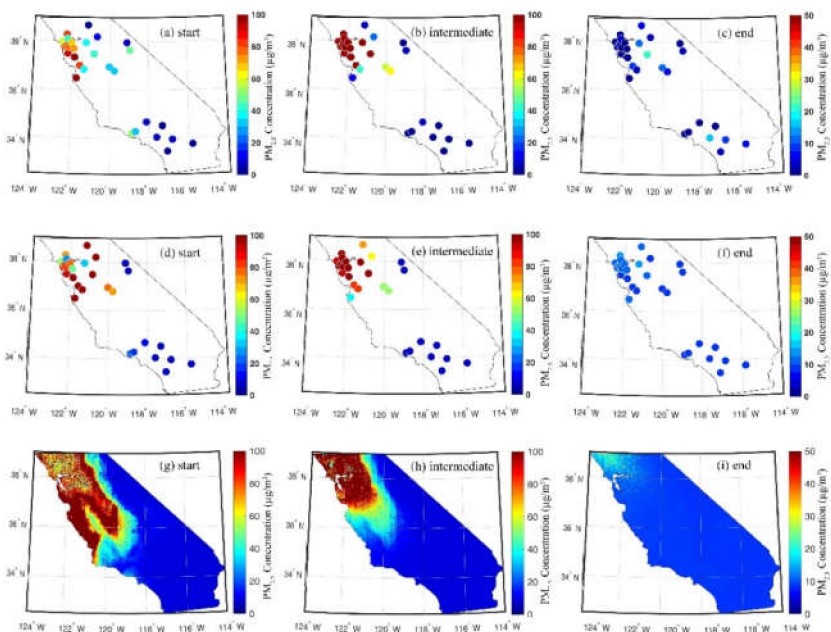

**Figure 10.** Distribution of PM2.5 concentration at 18:00 UTC peak. (**a**–**c**) Ground observed PM2.5 concentration; (**d**–**f**) Estimated PM2.5 concentration at the corresponding EPA sites based on Geoi-DBN model; (**g**–**i**) Spatial distribution map of estimated PM2.5 concentration based on Geoi-DBN.

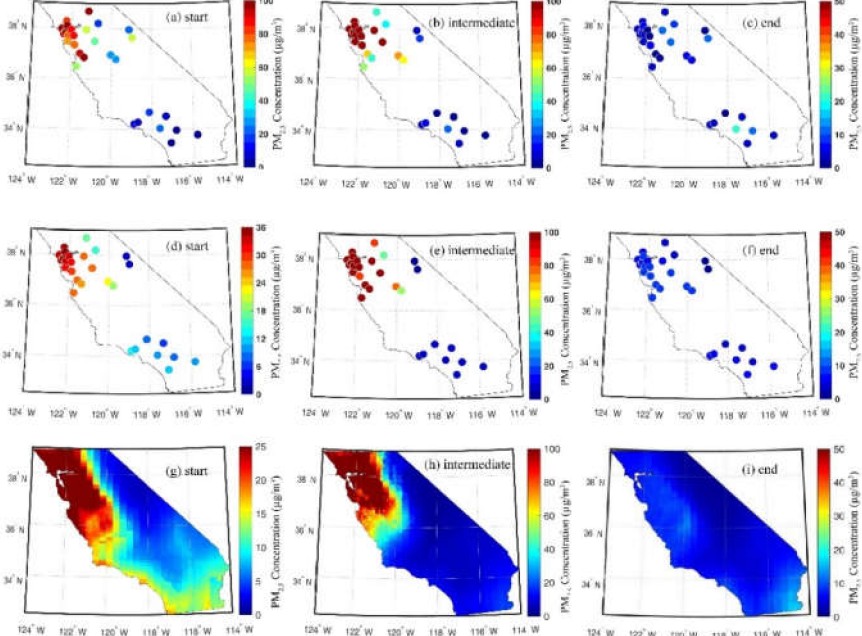

**Figure 11.** Distribution of PM2.5 concentration at 00:00 UTC peak. (**a**–**c**) Ground observed PM2.5 concentration; (**d**–**f**) Estimated PM2.5 concentration at the corresponding EPA sites based on the Geoi-DBN model; (**g**–**i**) Spatial distribution map of estimated PM2.5 concentration based on Geoi-DBN.

### 4.4. Uncertainty Analysis

The main goal of this paper was to predict the spatial distribution of PM2.5 for places with no ground monitors while leveraging the high resolution and full coverage of AOD data. Few studies have examined the relationship between ground PM2.5 and AOD at finer spatial and temporal resolution: essentially, the multi-source fused AOD, taking part

in estimation, still has uncertainty. Given these research gaps (downscaling, fusion, and prediction uncertainty), Figure 12 shows the curve of long-time series of estimated PM2.5 concentration (Figure 12a) and fused AOD (Figure 12b) as well as the histograms of their deviations from the ground observed data, respectively. Here, is shown the mean value of different AERONET sites (PM2.5 stations) and the matching up fused AOD (estimated PM2.5). The results showed that the fused AOD and the estimated PM2.5 concentrations fluctuated similarly, with lower values during the non-fire period and higher concentrations during the fire period. The difference between the bias of the AOD fusion product (from $-0.2$ to 0.4) and bias of PM2.5 estimation data (range of $-20$ µg m$^{-3}$ to 40 µg m$^{-3}$) was not distinct. The mean absolute errors were 0.063 (AOD) and 6.74 µg m$^{-3}$ (PM2.5), respectively. It should be noted that the error increased rather than decreased on some days during the wildfire outbreak, which may be caused by the absence of available observational data in its spatiotemporal neighborhood.

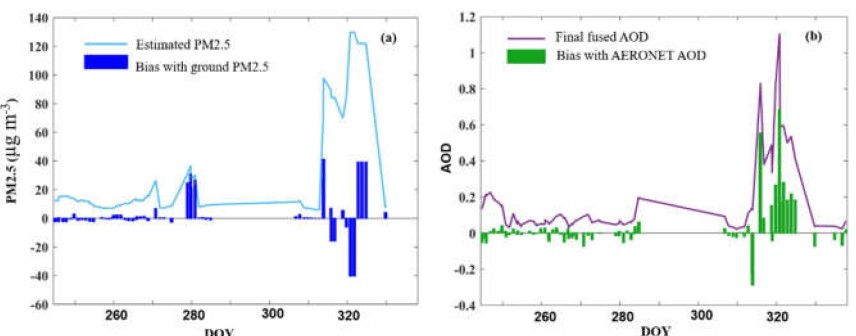

**Figure 12.** Variation curves of (**a**)PM2.5 and (**b**) AOD concentration and their bias with the ground observed data (DOY: Day of Year).

## 5. Conclusions

This paper has improved a novel machine learning model, namely the Geoi-DBN, considering the spatiotemporal autocorrelation with Gaussian kernel function to estimate the ground PM2.5 concentrations over central and south California during the wildfire period in the fall of 2018. In the estimation model, high spatial and temporal resolution AOD generated by fused AOD products and meteorological reanalysis data were used as the main input factors to improve the accuracy of the estimated ground PM2.5 concentration. The fusion of AOD data was carried out based on multi-scale and multi-source heterogeneous AOD data such as MERRA-2, GOES-16, MAIAC, and AERONET AOD products through the synergistic use of the MQQA-BME algorithm. The seamless AOD data generated by fusing the multi-source heterogeneous data could truly reflect the distribution characteristics of AOD. Compared with the AERONET AOD data, the cross-validated R$^2$ reached 0.69 and the RMSE was 0.072. The Geoi-DBN model could provide PM2.5 concentration data with high spatial and temporal resolution, which can meet the requirements of the timeliness and accuracy of monitoring data when catastrophic air pollution events such as fires occur. The Geoi-DBN algorithm was significantly better than the original DBN algorithm, and the cross-validated R$^2$ and RMSE were 0.86 and 6.84 µg m$^{-3}$, respectively.

Although the Geoi-DBN models performed better than the original DBN model in ground PM2.5 estimation, there are several ways to further improve the proposed models, which deserve further investigation. First, more input variables and longer archives can be used to further improve the models. Second, other sophisticated approaches such as deep learning could be utilized to refine the estimation accuracy for ground PM2.5 concentrations. The resolution of AOD and PM2.5 concentration can be improved in the near future with advanced sensor operation.

**Author Contributions:** X.W. designed the study. Q.C. collected and processed the data, analyzed the results, and wrote the original draft. F.Z. provided constructive comments on the paper. S.F., Y.M. and K.W. revised the paper. All authors have read and agreed to the published version of the manuscript.

**Funding:** This research was funded by the National Natural Science Foundation of China, grant number 42075125 and Shanghai Pujiang Program, grant number 20PJ1401800. The APC was sponsored by Shanghai Qi Zhi Institute.

**Data Availability Statement:** The data presented in this study are available on request from the corresponding author and first co-author.

**Acknowledgments:** The authors would also like to express gratitude to the data providers worldwide who provided the data/information used in this study.

**Conflicts of Interest:** The authors declare no conflict of interest.

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
