# Peer review of "High Spatiotemporal Resolution PM2.5 Concentration Estimation with Machine Learning Algorithm: A Case Study for Wildfire in California"

_remotesensing, doi:10.3390/rs14071635_

Round 1

Reviewer 1 Report

Complementing the deep learning method with information about the spatiotemporal correlations of the PM2.5 field is of interest. As shown by the authors, it really makes it possible to improve the resulting PM2.5 distribution fields. The manuscript should be published, but the work is framed very carelessly and needs to be improved. The language of the article requires significant revision, it is necessary to disclose all abbreviations. 

Reviewer 2 Report

Review of "Seamless Mapping of satellite-based ground PM2.5 Concentration with High Spatiotemporal Resolution"

I do not think that the title accurately captures the content of the paper.  To me, the title suggests that it is just the presentation of an algorithmic approach but the paper is really about the application to the wildfire case study.  Perhaps that could be indicated in a revised title.  Moreover, the word "ground" is sort of redundant in the title since PM2.5 is a ground-based measure.

Overall, I think that the approach/methodology presented here is quite good and should be published and I have tried to think of a different application where the methodology would work better.  But unfortunately, I have not been able to think of one. My primary problem with this work, detailed below, is that the data (shown in Figure 3) suggest a more complicated temporal behavior that is not fully captured with this methodology. Nonetheless, the methodology does provide an improvement and the data suggest further improvements are possible.

For the most part the paper is well written, but there are places where some of the phrasing is awkward.  For example, in the abstract (line 19) "...concentration under the confirm accuracy is still a grand challenge." would perhaps be better as "...concentration with the requisite accuracy is still a grand challenge."

Line 49: "...concentrations by satellite" should perhaps be "concentration from satellite"

Line 57: "...is large by using..." should perhaps be "...is large when using..."

Line 75: "...give a chance..." should perhaps be "provide a chance"

Line 82 "MODIS loaded" should perhaps be "MODIS flown"

Line 88, I'm not sure about the reference [7] here since that paper describes a merged aerosol product of data plus GOCART model and does not appear to be a reference for the MERRA-2 aerosol product.  Also the "A" is missing are the product is listed as MERR2 AOD data.

Since, I'm commenting of the MERRA-2 aerosol product, it is curious that you use the ECMWF reanalysis product for the meteorological variables and not the MERRA-2 reanalysis which, I think has all of the parameters you are considering.  Not that there is any real significance to the choice.

Lines 102 and 103: Since you are not directly commenting on the algorithm that is used, the sentence is awkward as written.

Line 205: do you mean to say that this can easily lead to model overfitting?

Could you possibly expand on the discussion of Figure 3, Lines 248-250? While I agree with your assertion that the spatial domain has an exponential dependence the temporal domain dependence seems more complicated with perhaps some sort of oscillatory behavior initially. While beyond the scope of this paper, there are some interesting studies being done on forest fire aerosol evolution combining MISR and aircraft (field measurements) that suggest a complex temporal evolution and an area ripe for future research (e.g.,  Junghenn Noyes, K. T., Kahn, R. A., Limbacher, J. A., and Li, Z.: Canadian and Alaskan wildfire smoke particle properties, their evolution, and controlling factors, from satellite observations, Atmos. Chem. Phys. Discuss. [preprint], https://doi.org/10.5194/acp-2021-863, in review, 2021.).

Line 315: "...it is difficult to account the estimation results..." missing the word for "it is difficult to account for the estimation results..."

Reviewer 3 Report

Standard English: there are a large number of variations in standard English which don't affect the meaning or readability, but may produce unfair unconcious judgement in the readers.  Some examples with corrections are below to demonstrate the point.  It is recommended that the paper be reviewed by someone who is intimately familiar with English: it is not the job of reviewers to correct all such grammatical errors.

line 19: "...under the confirm accuracy.."  grammatically this should be "confirmed" but I still don't understand the meaning.  Is this the algorithms stated accuracy?
line 36: "...becomes a generally concern..." => either "becomes a general concern" or "generally becomes a concern"
line 37: "...PM2.5 ground monitoring network and provide real time data," => "...PM2.5 ground monitoring networks that provide real time data,"
line 46: "Heterogeneous PM2.5 is caused by..." => "Heterogenity of PM2.5 is caused by..."

and so on throughout the paper.

Section 1: Introduction

Lines 52 and 63 are repetive, it would be better to cut off the second phrase that begins with "due to..." or replace with "due to the effects and variables discussed above."

line 120: why would the Moran index not represent the distribution of PM2.5?  State the assumptions being violated.

Section 3: Methods

section 3.1: The reader is not provided enough information about the MQQA or BME algorithms to understand why MQQA can't reconstruct missing data, why BME can be merged with probability theory, etc. More needs to be said about each of these algorithms which are to be merged.  The details can be left for the references, but the reader should not be compelled to go to the references just to follow the narrative.  Whether this additional information occurs in the methods or introduction is up to the authors, but the information IS needed. A paragraph would be sufficient: note how much information is provided about DBN.

Section 3.2: many of the variables are not defined.  Some are standard, but DOY, TP and perhaps PBLH should be defined.  Why is windspeed needed when you already have the components? Or is WS related to the weighting variables of the next section?

Figure 3: Given the scatter in the data, I have a hard time believing the sudden increase in temporal covariance at time = 0 based on a single point, unless it is fixed by assumption.  If so, this should be noted.

Results

lines 262 to 267: all these numbers are in the table, so should NOT be repeated in the text except in general statements such as "the R2 ranged from 0.3 to 0.53, with the fused R2 of 0.48 being acceptable due to gap filling."   Stating all numbers in the text simply serves to annoy the reader.

Equation 7: it should be noted that Reff is the area weighted average radius.

Discussion of figures 4,5: though the two diurnal peaks in AOD are noted, the PM only peaks later in the day: this should be mentioned - is it related to RH or other dynamics?  The PM is also more constant during the wildfire events than the AOD.  Any idea why?

Figure 7: I don't see that the CDF add anything, and are just a visual distraction that should be removed.  From my experience CDFs are useful for calculations only, but add nothing to understanding so should not be displayed.

Figure 8: yes, the figures are described in the caption, but could you label DBN and geoi-DBN rows so the reader can see immediately what is going on?  Perhaps the lower left corner of each map would be sufficient.

Figures 9,10: what do the three columns correspond to?  Different times? It's not explained in the caption, only in the text.  I would add "beginning, middle, end" to the bottome left corners of each map for the columns, accompanied by "observed, DBN-geoi, DBN-geoi extrapolated"

Reviewer 4 Report

Review of “Seamless Mapping of satellite-based ground PM2.5 Concentration with High Spatiotemporal Resolution” by Cui Q et al.

This manuscript by Cui Q et al. investigated a satellite-based ground PM2.5 concentrations mapping method. They utilized high spatiotemporal resolution AOD data and two machines learning techniques, Deep Belief Network (DBN) and Geoi-DBN. They demonstrated that the Geoi-DBN outperforms the original DBN method, by applying the methods to wildfire events that happened in California.

In general, this manuscript is reasonably written and organized. I recommend this manuscript for publication after some major revisions.

  1. Please, discuss why the Geoi-DBN method outperforms the DBN method.
  2. Although authors used wildfire events to investigate the surface PM mapping methods, important characteristics of wildfire events were not fully discussed. For example, during the wildfire events, surface PM concentrations are not always consistent with AOD observations, especially when fire smokes transport aloft. Please, provide deeper analyses and discussions on such cases. The biggest problems in the fire events are how to connect uplift smoke plume to surface concentrations.

L100

What is BME?

L212

What is TP?

Table 1

What is MAD?

Please, provide more information on the MPE. More spatial graphics will help.

L300

“It was found that 98%-99% relative humidity resulted in a 25% change 300 in AOD and 50%-80% relative humidity caused a 5% change in AOD [55].”

What does it mean? I don’t understand. Please, revise.

L330

“consistent with the diurnal pattern revealed in previous studies [59]”

need more explanation.

L337

“Results indicate that the spatial distribution of the two, with high values in the northwestern region and lower values in the southern region is consistent with the transport of smoke from wildfire according to HYSPLT online model”

Need more explanation. Please, show HYSPLIT analysis results.

L353

“to have a bearing on PM2.5 estimation”

What does it mean?

L361

“A large number of PBLHs samples are low values as seen in the distribution plot.”

What does this mean?

L364

“During these two time periods while the PBLH is low, temperature, wind speeds are also lower relative to their magnitudes at corresponding times during high PM2.5 concentration.”

I don’t think it was shown in the manuscript.

L391

“scatter plot of cross-validation of Geoi-DBN model”

It seems to be a simple scatter plot. I don’t understand why authors call it a cross-validation.

L378

“We found that the accuracy of Geoi-DBN algorithm was significantly better than the original DBN algorithm.”

Can you explain why?

L379

Figure 11b and 11d

Maybe figure numbers are wrong?

L385

“and the Geoi-DBN algorithm could estimate the ground PM2.5 concentration more accurately for introducing the priori information of observation PM2.5 concentration of surrounding stations.”

Need more explanation.

L437

Figure 11 is not a good way of showing model performance. Please, provide how data points were selected. Estimated PM2.5 should be compared with observed PM2.5 concentrations. The fused AOD should be compared with AERONET AOD. Please, provide actual observations, not biases.

What is “variation curves”? Do you mean temporal variation?

Explain DOY.

L433

“It should be noted that the error increased rather than decreased on some days during the wildfire outbreak, which may be caused by the absence of available observational data in its spatiotemporal neighborhood.”

Well, it can be due to your model.

L462

“The synergistic use of Sentinel-like AOD data is likely improve the resolution to meter-level in AOD fusion and seamless mapping ground PM2.5 concentration in the near future.”

What do you mean?

Round 2

Reviewer 4 Report

I appreciate authors' efforts to revise the manuscript.